# BENCHMARKING OPEN-SET RECOGNITION BEYOND VISION-LANGUAGE PRE-TRAINING

## ABSTRACT

Vision-language models (VLMs) with open-vocabulary pre-training can still fail in classification tasks, especially when the granularity of downstream labels misaligns with the supervision during pre-training. In such cases, a few-shot training set is necessary to define the classification task on demand. Motivated by this, we investigate the performance of VLMs in open-set recognition (OSR), where a VLM is fine-tuned on a few-shot training set to recognize closed-set classes while identifying and rejecting samples from open-set ones, *i.e.*, without compromising its open-vocabulary capabilities. We design a comprehensive benchmark to study this problem, varying along four key axes: (1) label granularity (fine- vs. coarse-grained classes), (2) the semantic distance of open-set classes from closed-set ones (OSR hardness), constructed using hierarchical taxonomies, (3) the number of training samples per class, and (4) fine-tuning objectives (discriminative vs. likelihood-based). Through systematic evaluation of CLIP-based and diffusion-based VLMs, we find that discriminative approaches are often misaligned with standard OSR hardness metrics, leading to unreliable rejection behavior. In contrast, the likelihood-based paradigm becomes tractable in the context of VLMs, and we propose a simple method based on a likelihood-ratio test that achieves strong OSR performance when given sufficient examples to model class-conditional likelihoods. Overall, our results demonstrate that OSR remains a relevant and underexplored challenge even in the era of VLMs. We provide actionable insights and a new benchmark to support future research toward more robust open-world recognition.

## 1 INTRODUCTION

With the advent of open-vocabulary pre-training and the emergence of vision-language models (VLMs) (Ilharco et al., 2021; Rombach et al., 2022), one might naturally question whether training a dedicated open-set recognition (OSR) model is still necessary—after all, open-vocabulary models appear to be inherently open-set. For example, in the zero-shot classification paradigm of CLIP (Radford et al., 2021), each class is represented by a natural language prompt (*e.g.*, "a photo of an aircraft"), and the model classifies an image by selecting the prompt with the highest image-text similarity. If the model has encountered relevant visual concepts during pre-training and semantically understands the prompts, it should, in principle, be able to determine whether an image belongs to predefined closed-set classes $\mathcal{C}$, satisfying the core objective of OSR: to distinguish between known and unknown inputs.

We argue that OSR remains highly relevant in the era of VLMs. We highlight an underexplored limitation: although VLMs are exposed to diverse visual concepts during pre-training, their understanding is fundamentally constrained by the semantic granularity of the accompanying captions. For example, while many aircraft images may appear in the pre-training data, they are typically labeled at a coarse-grained level such as "an image of an aircraft", since fine-grained annotations like specific aircraft models require expert-level knowledge and are rarely available at scale. As a result, when the granularity of downstream labels misaligns with pre-training supervision, the zero-shot classification paradigm breaks down, *e.g.*, failing to distinguish fine-grained aircraft variants. In such cases, a few-shot training set becomes essential to define the classification task on demand. While recent efforts have explored fine-tuning VLMs using few-shot examples for such tasks (Yue et al., 2024), it remains unclear whether this adaptation compromises the open-vocabulary gener-

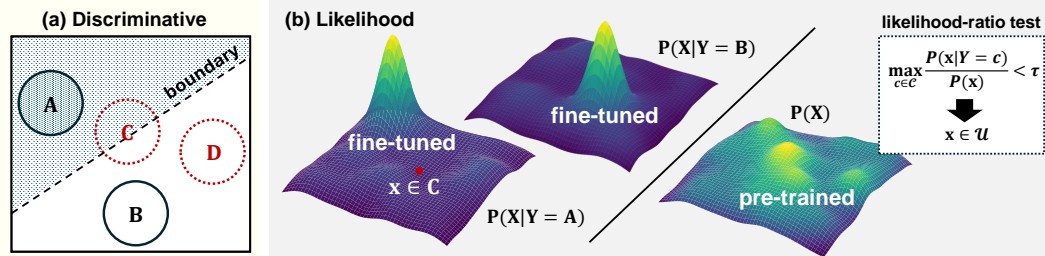

Figure 1: A conceptual example of (a) discriminative approach and (b) likelihood approach in OSR. On the left, we show an example in a 2-D sample space, where $\{A, B\}$: closed-set classes, $\{C, D\}$: open-set classes, and dotted line: a linear classification boundary trained on $A, B$. On the right, we plot conditional and marginal likelihood over the sample space. Explanations are in the main text.

alization that makes VLMs powerful to begin with, and potentially degrades their performance in OSR settings.

To systematically investigate how fine-tuning VLMs on closed-set classes $\mathcal{C}$ affects their ability to reject samples from open-set classes $\mathcal{U}$, we introduce **OpenVL-Bench**—a comprehensive *Open*-set benchmark to test *V*ision-*L*anguage models on tasks beyond their pre-training knowledge—which comprises *60* challenging OSR tasks spanning multiple domains, organized along four key axes:

- **Label granularity**. We focus on real-world scenarios where the zero-shot paradigm of VLMs is limited, *i.e.*, tasks where labeling requires expert-level knowledge. These tasks range from fine-grained classification (*e.g.*, aircraft model recognition, plant disease diagnosis, industrial defect detection) to coarse-grained species taxonomy prediction under Linnaean naming.
- **OSR hardness**. Next, we leverage hierarchical structure within each dataset to construct OSR splits with varying distances between known classes $\mathcal{K}$ and unknown classes $\mathcal{U}$ (*i.e.*, hardness), inspired by similar practices in the OSR literature (Vaze et al., 2021; Yue et al., 2024).
- **Sample size**. We vary labeled training examples per class from 1 to 16 to study sample efficiency.
- **Open-set score**. Importantly, OSR methods rely on a paradigm to compute open-set score, where samples with low score are rejected as from $\mathcal{U}$. We evaluate two common paradigms to compute open-set score. The first is likelihood-based, which estimates $P(X \mid Y)$ and rejects inputs with low maximum likelihood (Yoshihashi et al., 2019; Sun et al., 2020). The second is the more prevalent discriminative paradigm, which learns $P(Y \mid X)$ and rejects inputs whose maximum predicted probability falls below a threshold (Bendale & Boult, 2016; Ge et al., 2017). This is widely adopted in traditional OSR settings without VLMs (Vaze et al., 2021; Lang et al., 2024).

We benchmark two main lines of VLMs—CLIP-based models trained via contrastive alignment on image-text pairs (Radford et al., 2021; Ilharco et al., 2021), and diffusion-based models trained to approximate the image likelihood conditioned on textual prompts (Rombach et al., 2022). This leads to the following findings:

1. The discriminative approach is inherently misaligned with standard OSR hardness metrics (Fu et al., 2020), as conceptually illustrated in Figure 1a. For example, the task is considered harder when the open-set class is $\mathcal{U} = \{C\}$ rather than $\mathcal{U} = \{D\}$, since $C$ is visually similar to the closed-set classes $A$ and $B$. Paradoxically, a discriminative classifier trained only to separate $A$ and $B$ may perform better in the harder case $\mathcal{U} = \{C\}$, simply because $C$, being near the decision boundary, is more likely to be rejected due to low confidence. This highlights a key limitation of discriminative models: there is no guarantee that a decision boundary trained exclusively on closed-set classes will be near all potential open-set classes (*e.g.*, $C$ or $D$).
2. The likelihood-based paradigm becomes tractable and effective in the context of VLMs. Traditional likelihood-based methods often struggle in OSR, as they estimate $P(X|Y)$ using only known classes, with no guarantee of extrapolating to unknown ones. However, this limitation is substantially mitigated in the VLM setting. Since pre-training exposes VLMs to a broad range of visual concepts, it learns a much more accurate marginal distribution $P(X)$, even for samples from open-set classes. As a result, if we can accurately model $P(X|Y = A)$ and $P(X|Y = B)$ in fine-tuning, a sample $\mathbf{x}$ from $\mathcal{U}$ (*e.g.*, $C$ or $D$) is naturally assigned lower likelihood (Figure 1b), because it has anchored $A$ and $B$ around semantically meaningful modes, and recognizes when $\mathbf{x}$ does not resemble either.

Figure 2: Paradigms of three fine-tuning approaches to enable OSR with VLMs. (a) CLIP-Adapter-based methods. (b) Prompt-tuning-based methods. (c) A diffusion-based method.

3. For diffusion-based models, we propose a simple method **SD-LRT** that approximates a likelihood-ratio test (LRT) illustrated in Figure 1b. Grounded in Neyman–Pearson optimality (Hastie et al., 2009), we find it delivers strong empirical results. These findings suggest that (a) likelihood-driven OSR deserves renewed attention and refinement, and (b) future research should either advance such generative approaches or devise ways to overcome the intrinsic limitations of purely discriminative methods.

## 2 PROBLEM FORMULATION

### 2.1 OPEN-SET RECOGNITION

The goal of open-set recognition (OSR) is to accurately classify inputs from a set of closed-set classes $\mathcal{C}$, while identifying and rejecting inputs from open-set classes $\mathcal{U}$ that are not observed during fine-tuning. Importantly, $\mathcal{C}$ is disjoint from $\mathcal{U}$. We assume access to a pre-trained vision-language model (VLM), which takes an image $\mathbf{x}$ and a text prompt $y_c$ that describes $c \in \mathcal{C}$ as input, and outputs the similarity between $\mathbf{x}$ and $c$. The training set for each $c \in \mathcal{C}$ consists of a small number of $N$ labeled examples $\{\mathbf{x}_i, y_c\}_{i=1}^{N}$, which is used to fine-tune the VLM. Given a test image $\mathbf{x}$, the model rejects it as from $\mathcal{U}$ if its *open-set score* $s(\mathbf{x})$ (introduced later) is smaller than a threshold $\tau$. If $\mathbf{x}$ is not rejected, it is classified as $c$ that has the maximum similarity with $\mathbf{x}$.

Next, we introduce two types of VLMs, CLIP-based in Section 2.2 and diffusion-based in Section 2.3 and explain how they are fine-tuned and how their open-set scores are computed.

### 2.2 OSR FOR CLIP-BASED VLMS

CLIP-based models consist of a visual encoder $\mathbf{V}$ and text encoder $\mathbf{T}$ that maps each image $\mathbf{x}$ and prompt $y$ to an aligned feature space in $\mathbb{R}^d$. While these models are pre-trained in a discriminative fashion by aligning the image and text features on paired dataset, we can fine-tune them with both the discriminative and likelihood objectives.

**Discriminative objective**. This corresponds to a line of work called CLIP-Adapter Gao et al. (2024); Zhang et al. (2021), which is illustrated in Figure 2a. The key idea is to append a trainable network parameterized by $\theta$ to $\mathbf{V}$, where we denote the adapted visual encoder as $\mathbf{V}_\theta$. Then we learn $\theta$ with a discriminative objective:

$$\min_\theta \sum_{c \in \mathcal{C}} \sum_{i=1}^{N} \left[ \mathcal{L}\Big( \mathbf{T}(y_\mathcal{C})^\intercal \big( \alpha \mathbf{V}_\theta(\mathbf{x}_i) + \mathbf{V}(\mathbf{x}_i) \big), c \Big) \right], \tag{1}$$

where $\mathbf{T}(y_\mathcal{C}) \in \mathbb{R}^{d \times |\mathcal{C}|}$ denotes the text feature matrix for all closed-set classes, $\mathcal{L}$ denotes the cross-entropy loss (with softmax over logits absorbed in), and $\alpha$ is a balancing hyper-parameter. Here, $\mathbf{T}(y_\mathcal{C})$ essentially defines a classification boundary of a linear classifier. After training, we use a discriminative open-set score defined below:

$$s_{\text{dis}}(\mathbf{x}) = \max_{c \in \mathcal{C}} \mathbf{T}(y_c)^\intercal \big( \alpha \mathbf{V}_\theta(\mathbf{x}) + \mathbf{V}(\mathbf{x}) \big), \tag{2}$$

which effectively rejects $\mathbf{x}$ if its most confident classification logit is smaller than $\tau$.

**Likelihood-based score**. We do not consider a likelihood-based training objective for CLIP, as it requires model adaptation (Ramesh et al., 2022). We focus on prompt-tuning methods (Zhou et al.,

2022b;a) that enable a likelihood-based open-set score (Figure 2b). Instead of explicitly constructing a classifier, these methods learn an adapted text encoder $\mathbf{T}_\phi$ that produces class-specific prototypes, enabling classification through feature similarity rather than learned decision boundaries:

$$\min_\phi \sum_{c \in \mathcal{C}} \sum_{i=1}^{N} \mathcal{L} \left( \left\{ \langle \mathbf{V}(\mathbf{x}_i), \mathbf{T}_\phi(y_{\bar{c}}) \rangle \right\}_{\bar{c} \in \mathcal{C}}, c \right), \tag{3}$$

where $\langle \cdot, \cdot \rangle$ denotes the cosine similarity. Then we define a likelihood-based score given by:

$$s_{\text{proto}}(\mathbf{x}) = \max_{c \in \mathcal{C}} \langle \mathbf{V}(\mathbf{x}), \mathbf{T}_\phi(y_c) \rangle, \tag{4}$$

which rejects $\mathbf{x}$ if its similarity with the closest prototype is smaller than $\tau$. Using the example in Figure 1a, if we learn a prototype for features in $A$ and $B$, respectively, using $s_{\text{proto}}$ in Eq. (4) no longer has the limitation from using a decision boundary.

## 2.3 OSR FOR DIFFUSION-BASED VLMS

A diffusion-based VLM is a generative model with a denoising network $\mathbf{D}$. Given a noisy image $\mathbf{x}_t$ sampled from the forward diffusion path $q(\mathbf{x}_t|\mathbf{x})$, the output of $\mathbf{D}$ reconstructs $\mathbf{x}$, where $t \in [0, \ldots, T]$ is diffusion time-step that controls the noise level. However, as the diffusion formulation has no explicit encoders that transform samples into feature vectors, it is difficult to apply the discriminative objective. Hence we only consider a likelihood objective, as illustrated in Figure 2c.

**Likelihood objective**. Following (Yue et al., 2020), we learn a set of class-specific low-rank (LoRA) adaptation parameters (Hu et al., 2022) $\{\theta_1, \ldots, \theta_{|\mathcal{C}|}\}$ for $\mathbf{D}$, which are trained to maximize the class-conditional likelihood $P(X|Y)$ for each $Y = c \in \mathcal{C}$:

$$\min_{\theta_1, \ldots, \theta_{|\mathcal{C}|}} \sum_{c \in \mathcal{C}} \sum_{i=1}^{N} \mathcal{L}_{\text{DM}}(\mathbf{D}(\cdot; \theta_c), \mathbf{x}_i, y_c), \tag{5}$$

$$\text{where } \mathcal{L}_{\text{DM}}(\mathbf{D}, \mathbf{x}, y) = \sum_{t=1}^{T} \mathbb{E}_{\mathbf{x}_t \sim q(\mathbf{x}_t|\mathbf{x})} \left[ \|\mathbf{x} - \mathbf{D}(\mathbf{x}_t, y, t)\|^2 \right], \tag{6}$$

where $\mathbf{D}(\cdot; \theta_c)$ denotes the model adapted by $\theta_c$ through LoRA injection, and we prove in appendix that $-\mathcal{L}_{\text{DM}}(\mathbf{D}(\cdot; \theta_c), \mathbf{x}, y_c)$ is the lower bound of log-likelihood $\log P(X = \mathbf{x}|Y = c)$.

**Likelihood-ratio test (LRT)**. After training with Eq. (6), we propose to use the following open-set score, which approximates the log-likelihood-ratio test.

$$s_{\text{lrt}}(\mathbf{x}) = \min_{c \in \mathcal{C}}[\mathcal{L}_{\text{DM}}(\mathbf{D}(\cdot; \theta_c), \mathbf{x}, y_c) - \mathcal{L}_{\text{DM}}(\mathbf{D}, \mathbf{x}, \hat{y})] \approx \max_{c \in \mathcal{C}} \log \frac{P(\mathbf{x}|Y = c)}{P(\mathbf{x})}, \tag{7}$$

where $\hat{y}$ denotes an empty prompt, and $\mathcal{L}_{\text{DM}}(\mathbf{D}, \mathbf{x}, \hat{y})$ corresponds to an unconditional, marginal likelihood. Rejecting $\mathbf{x}$ when $s_{\text{lrt}} < \tau$ thus implements a likelihood-ratio test, which is the optimal decision rule (in the Neyman–Pearson sense) for deciding between the two hypotheses: $H_0 : \mathbf{x} \in \mathcal{C}$ and $H_1 : \mathbf{x} \notin \mathcal{C}$, *i.e.*, deciding if $\mathbf{x}$ should be rejected. In this work, we use Stable Diffusion (Rombach et al., 2022) as $\mathbf{D}$, and hence we term this approach **SD-LRT**. We show in Section 5.2 that SD-LRT achieves strong empirical performance in practical OSR scenarios.

## 3 OPENVL-BENCH

Recall from introduction that OpenVL-Bench is organized along four axes. We have covered the fine-tuning objectives in the previous section. In Section 3.1, we detail how our OSR tasks are constructed by varying label granularity as well as similarity between $\mathcal{C}$ and $\mathcal{U}$. In Section 3.2, we introduce the training sample sizes, evaluation protocols and other implementation details.

### 3.1 OSR TASKS

To construct meaningful open-set tasks, we begin by selecting datasets with practical real-world tasks, that are insufficiently addressed by the zero-shot classification paradigm of current VLMs.

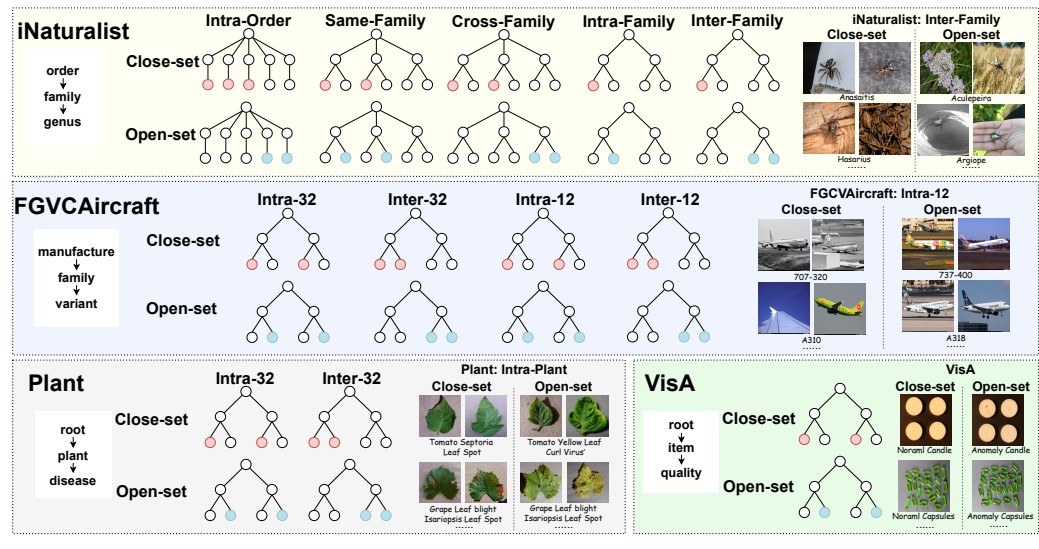

Figure 3: Overview of our OpenVL-Bench. For each setting, we run experiments with training examples $N = 1, 2, 4, 8, 16$ for each class across 5 random seeds.

We discuss our selection process and include the zero-shot classification results in Appendix. This leads to 4 datasets covering a diverse range of tasks and label granularity. We then design settings by constructing data splits on each of them to systematically assess the OSR performance of VLMs.

**FGVCAircraft (Maji et al., 2013)**. It contains images from 100 aircraft variants, organized in a hierarchical taxonomy: manufacturer (*e.g.*, Airbus, Boeing) at the top, family (*e.g.*, 707, 737) in the middle, and variant (*e.g.*, 707-320, 737-400) at the bottom. This dataset aligns well with the characteristics of OSR in the VLM era, where the model may have seen the visual concept but not its precise, domain-specific label (aircraft variants).

- We design two types of evaluation settings—**Intra-Family** and **Inter-Family**—to study how semantic distance affects the OSR performance of VLMs. In the *Intra-Family* settings, $\mathcal{U}$ are sampled from the same families as the closed-set classes. This implies a smaller semantic gap between known and unknown classes, making the OSR task more challenging. In contrast, the *Inter-Family* settings select $\mathcal{U}$ from different families than the closed-set ones, resulting in a larger semantic distance and hence a relatively easier open-set scenario.
- For each of the two types of settings on FGVCAircraft dataset, we study two cases where $|\mathcal{U}|$ is 12 or 32 (*e.g.*, Intra-12, Intra-32). In our design, increasing the number of $\mathcal{U}$ reduces the number of closed-set classes (as the total class number is fixed at 100), which also reduces the amount of available supervision during training. So, we expect *Intra-32* and *Inter-32* to be the most challenging configurations, due to both increased open-set size and reduced supervised information.

**iNaturalist (Van Horn et al., 2018)**. It contains 675,170 training and validation images from 5,089 natural fine-grained categories. All samples are classified according to kingdom, phylum, class, order, family, genus, and species, which represent the natural semantic relationships between different categories. We select a subset and adopt the genus level as the classification target, which can help zoologists group closely related species to better understand their evolutionary relationships and shared traits. This choice is also motivated by the observation that during VLM pre-training, animal images are often associated with their common or species-level names (Parashar et al., 2023).

- As shown in Figure 3, we first define the **Intra-Order** setting, where all classes are sampled from different families within the same order. This results in family-level semantic distances between classes, making both closed-set and OSR tasks comparatively easier.
- Second, we introduce a pair of comparative settings: **Same-Family** and **Cross-Family**. In both settings, $\mathcal{C}$ is identical. In *Same-Family*, $\mathcal{U}$ is sampled from the same families as $\mathcal{C}$; in *Cross-Family*, $\mathcal{U}$ come from different families within the same order. This design allows us to isolate the effect of open-set variation by keeping the closed-set fixed. In particular, *Same-Family* corresponds to a hard OSR task according to the standard definition (Vaze et al., 2021; Dhamija et al.,

2018), due to strong visual similarity between $\mathcal{C}$ and $\mathcal{U}$. whereas *Cross-Family* poses a different, underexplored challenge: knowledge learned from $\mathcal{C}$ does not easily generalize to semantically distant open-set inputs (from different families).

- Lastly, we construct another comparative pair with the same $\mathcal{C}$: **Intra-Family** and **Inter-Family**, which poses two distinct challenges similar to the previous pair, but made more extreme. *Intra-Family* vs. *Same-Family*: even stronger similarity between $\mathcal{C}$ and $\mathcal{U}$ by having them come from the same family; *Inter-Family* vs. *Cross-Family*: even harder to generalize, as $\mathcal{C}$ comes from closely related genera in the same family, instead of diverse genera from multiple families.

**Plant Diseases Dataset (Mohanty et al., 2016)**. The dataset has 38 classes for leaf disease classification and contains 14 leaf categories. Each leaf category contains one healthy class and several disease classes. Hence, this dataset includes composite labels that combine plant species (*e.g.*, "Grape Leaf") with specific diseases (*e.g.*, "Blight Isariopsis Leaf Spot"). Such composite labels are rarely encountered during VLM pre-training, making the dataset a suitable candidate for evaluating the OSR capabilities of VLMs.

- Similar to previous benchmarks, we define two evaluation settings—**Intra-Plant** and **Inter-Plant**—to analyze the impact of semantic distance. In the *Intra-Plant* setting, the closed-set and open-set samples may come from the same plant species, *e.g.*, different diseases affecting the same type of leaf, making the open-set task more difficult. In contrast, *Inter-Plant* assumes $\mathcal{U}$ originate from plant types unseen in the closed-set, leading to a larger semantic gap and an easier recognition scenario.

**VisA (Zou et al., 2022)**. It comprises high-resolution images (approximately 1.5K x 1K pixels) spanning 12 object categories, each containing both defect-free (normal) and defective (anomalous) industrial parts. Industrial anomaly detection often requires identifying subtle defects in objects when only "good" samples are available during training (Bergmann et al., 2019; Roth et al., 2022). This is because defective samples are rare, diverse, or costly to annotate in real-world industrial settings, making it impractical to collect comprehensive labeled datasets covering all possible failure modes. Hence the task can be viewed as a special case of OSR, where $\mathcal{C}$ are categories of industrial parts without defects, and $\mathcal{U}$ are faulty parts. In the context of VLMs, upstream pre-training is not expected to captures downstream-specific semantics such as what constitutes "good" or "bad" industrial parts, further necessitating the need of downstream OSR training. This makes the task extremely fine-grained, as the distinction between normal and defective parts often hinges on subtle, localized visual cues that are difficult to detect even by humans.

The four datasets we select are both challenging for VLMs and aligned with the definition of OSR in the VLM era. For each dataset, we construct multiple evaluation settings to systematically assess the OSR capabilities of VLMs.

## 3.2 EVALUATION SETTINGS

**Number of training samples**. We selected 1-, 2-, 4-, 8-, and 16-shots as the training data size, which is a common setting when fine-tuning VLMs (Zhou et al., 2022a; Khattak et al., 2023; Zhang et al., 2021). Given that VLMs provide strong priors, few-shot evaluation is not only realistic but also a natural test of their adaptability with minimal supervision.

**Evaluation metrics**. On the FGVCAircraft, iNaturalist and Plant with disease, we followed (Vaze et al., 2021; Sun et al., 2021; Fu et al., 2020) using the threshold-free area under the Receiver-Operator curve (AUROC) as the evaluation metric for OSR and macro $F_1$-score for closed-set classification. On VisA dataset, we used image-level AUROC and $F_1$-score at optimal threshold ($F_1$-max) following (Jeong et al., 2023; Defard et al., 2021).

**Compared approaches**. For CLIP-based methods using likelihood-based score, we compare popular prompt-tuning methods: CoCoOp (Zhou et al., 2022a), MaPLe (Khattak et al., 2023) and LoCoOp (Miyai et al., 2023). For CLIP with discriminative objective, we compare Tip-Adapter and Tip-Adapter-F (Zhang et al., 2021). Note that "-F" denotes a stronger variant of Tip-Adapter that introduces additional training parameters. Also, we test the diffusion-based SD-LRT mentioned in Sec. 2. On VisA, we used SPADE (Cohen & Hoshen, 2020), PaDiM (Defard et al., 2021), PatchCore (Roth et al., 2022), WinCLIP (Jeong et al., 2023) and VAND (Chen et al., 2023).

**Implementation Details**. We used SD 2.0 following (Li et al., 2023; Yue et al., 2024). For LoRA matrices rank, we used 16 for each dataset. We use "a photo of [c], a type of [SC]" for CLIP based

baseline methods on FGVCAircraft, where [c] denotes the name of class $c$ and [SC] denotes the dataset-specific super-class name. We used the same prompt as the original papers for the anomaly detection methods. For our method, we used "a photo of [c], a type of [SC]" for both SD models with and without LoRA. All experiments were conducted on a single NVIDIA A100 GPU.

# 4 RELATED WORK

**Open-Set Recognition and Few-Shot Learning**. OSR was first formalized in Scheirer et al. (2013), with Bendale & Boult (2016) introducing deep learning approaches. A key insight is that open-set performance correlates positively with closed-set performance (Vaze et al., 2021). Recent advances address fine-grained shifts (Lang et al., 2024) and few-shot scenarios through transformation consistency (Jeong et al., 2021). FSOSR approaches include negative prototype generation (Zhang et al., 2024), background-as-unknowns strategies (Song et al., 2022), and contextual transductive learning (Wu et al., 2025).

**VLM-based Open-Set Methods**. Vision-language models enable new OSR approaches through prompt engineering and fine-tuning. CLIP-based methods adapt threshold-based detection (Khattak et al., 2023), while prompt-tuning approaches like CLIPN (Wang et al., 2023) focus on OOD detection. Recent works propose anti-associative prompt tuning (Ren et al., 2024), negative prompts (Nie et al., 2024), and large model collaboration (Qu et al., 2023). However, Miller et al. (2024) demonstrate that VLMs still make closed-set assumptions and degrade under open-set conditions. Few-shot VLM adaptation has evolved through parameter-efficient methods like LoRA (Zanella & Ben Ayed, 2024) and two-stage frameworks (Farina et al., 2025).

# 5 RESULTS

## 5.1 CLIP-BASED APPROACHES

**CLIP-based approaches are sample-efficient.** CLIP-based approaches generally perform well on closed-set classification. Prompt-tuning approaches (e.g., CoCoOp, MaPLe) learn class-specific prototypes using only a few labeled examples, making them highly sample-efficient. We hypothesize that this is because of their stronger zero-shot performance compared to diffusion-based SD-LRT. They also benefit from additional supervision and can show strong improvements in closed-set performance as shots increase. For example, in the Intra-Plant setting (Table 1), Tip-Adapter-F improves from a closed-set $F_1$-score of 45.0 (1-shot) to 85.4 (16-shot). However, it is clear from Figure 5 and Figure 6 that likelihood-based approach using a generative model (SD-LRT) often surpass CLIP-based approaches as training sample size increases.

**Meaningful decision boundary is not guaranteed.** Despite running multiple seeds, we observe from the plots that the open-set metric of CLIP-based approaches is often less stable as sample size increases, compared to their steady increases in the closed-set metric. Additionally, the learned decision boundary often suffers

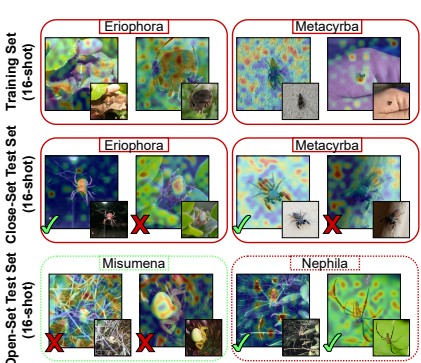

Figure 4: Visualizing the classification cue used by Tip-Adapter-F with Grad-CAM (Selvaraju et al., 2017). We display the attention map on 2 classes in 16-shot Cross-Family setting. Border colors indicate semantic relationships (same color for same superclass, different colors for different superclasses).

from generalization problems, *e.g.*, when it becomes more difficult to generalize the closed-set knowledge to open-set (Inter-Family vs. Cross-Family in Figure 6), we observe a performance drop for most CLIP-based approaches, despite $\mathcal{U}$ in fact having a larger semantic distance from $\mathcal{C}$ (*c.f.* improvements of SD-LRT across the two settings). We show qualitative examples in Figure 4 where the learned classification cue can be highly correlated with image backgrounds. This provides empirical evidence for the problems highlighted in Figure 1a. We also observe that the likelihood-based SD-LRT suffers less from this problem, often exhibiting consistent improvements in closed-set and

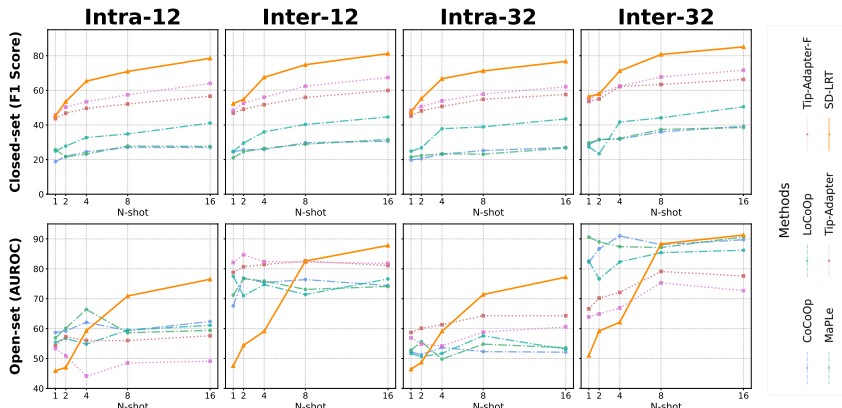

Figure 5: Performance comparison on FGVCAircraft dataset under four settings combining different open-set ratios (12 or 32 open-set classes against 88/68 closed-set classes) and semantic relationships (Intra/Inter-family).

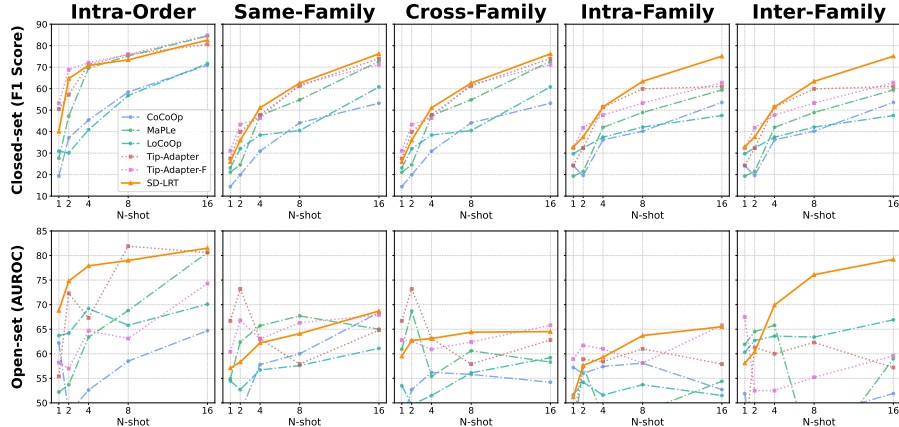

Figure 6: Performance comparison on iNaturalist dataset across five settings. We use $F_1$-score as the evaluation metric for closed-set classification, and AUROC as the metric for OSR.

open-set metric, which aligns with previous analysis. We also find that prompt-tuning methods, while underperforming adapter-based approaches on closed-set tasks, can achieve better open-set performance in certain scenarios (*e.g.*Inter-32 and Intra-32 in Figure 5).

## 5.2 DIFFUSION-BASED SD-LRT

**It achieves strong results in open-set evaluation.** In addition to closed-set performance, SD-LRT demonstrates a clear and stable upward trend in open-set AUROC as shots increase. For instance, in Figure 5 under the *Intra-32* setting, it achieves **77.2** AUROC (16-shot), outperforming Tip-Adapter-F (**64.3**) and CoCoOp (**53.6**). Similarly, in Table 1 (*Intra-Plant*), SD-LRT reaches **81.5** AUROC, significantly higher than Tip-Adapter-F (**72.6**) and LoCoOp (**72.1**). This strongly supports our analysis on the advantage of likelihood-based methods.

**It is also a strong closed-set classifier.** SD-LRT shows reliable performance gains on closed-set classification as the number of labeled samples increases and often significantly outperforms CLIP-based approaches in the 4-, 8-, and 16-shot regimes, *e.g.*, SD-LRT achieves **92.1** AUROC in the *Inter-Plant* setting (16-shot) and **85.1** in the *Inter-32* setting (16-shot), outperforming all other methods. This is consistent with previous findings (Yue et al., 2024), where generative pre-training was shown to retain fine-grained image attributes that are useful for downstream classification, demonstrating the potential of generative VLMs for downstream classification.

| Methods | Intra-Plant | | | | | Inter-Plant | | | | |
|---|---|---|---|---|---|---|---|---|---|---|
| | 1 | 2 | 4 | 8 | 16 | 1 | 2 | 4 | 8 | 16 |
| CoCoOp | 32.9 / 40.7 | 38.7 / 44.1 | 38.5 / 60.1 | 46.5 / 58.8 | 56.7 / 54.7 | 32.3 / 42.4 | 43.4 / 43.2 | 51.8 / 48.6 | 56.9 / 54.9 | 68.9 / 69.1 |
| MaPLe | 34.7 / 47.6 | **61.5** / 48.3 | 63.5 / 51.6 | 76.1 / 50.2 | 85.6 / 66.4 | 49.8 / 48.9 | 62.0 / 67.9 | 68.1 / 64.8 | 82.4 / 70.3 | 88.8 / 78.1 |
| LoCoOp | 22.6 / 53.1 | 25.4 / 56.9 | 42.2 / 55.1 | 58.0 / 59.3 | 72.9 / 72.1 | 22.6 / 60.6 | 35.5 / 57.0 | 40.5 / 62.1 | 55.9 / 67.8 | 71.6 / 70.9 |
| Tip-Adapter | 46.2 / 56.2 | 60.3 / **62.4** | 73.2 / 67.1 | 76.3 / 67.8 | 80.8 / 67.6 | 51.1 / 69.5 | 65.0 / 78.8 | 82.4 / 79.3 | 85.2 / 79.3 | 88.1 / 83.5 |
| Tip-Adapter-F | 45.0 / **64.7** | 47.4 / 58.0 | 64.9 / 62.2 | 71.2 / 65.2 | 85.4 / 72.6 | **54.8** / **69.9** | 54.9 / 68.7 | 79.2 / 73.7 | 81.3 / 82.5 | 92.1 / 87.0 |
| SD-LRT | **52.1** / 51.0 | 54.7 / 57.9 | **74.5** / **71.3** | **81.2** / **76.2** | **89.2** / **81.5** | 45.2 / 64.9 | **65.1** / **83.2** | **83.5** / **87.0** | 86.5 / **89.0** | 92.1 / **94.7** |

Table 1: Performance comparison across plant disease datasets under two settings: Intra-Plant and Inter-Plant. Results are reported as $F_1$-score / AUROC (%). SD-LRT exhibits progressively superior performance with increasing shot numbers, demonstrating strong feature learning capabilities as visualized in Figure 7.

| Methods | VisA | | |
|---|---|---|---|
| | 1 | 2 | 4 |
| SPADE | 80.7 / 79.5 | 81.7 / 80.7 | 82.1 / 81.7 |
| PaDiM | 75.3 / 62.8 | 75.7 / 67.4 | 78.0 / 72.8 |
| PatchCore | 81.7 / 79.9 | 82.5 / 81.6 | 84.3 / 85.3 |
| Winclip | 83.1 / 83.8 | 83.0 / 84.6 | 84.2 / 87.3 |
| VAND | 86.9 / 91.2 | 87.7 / **92.2** | 88.4 / 92.6 |
| SD-LRT | **88.1** / **91.3** | **89.3** / 92.1 | **91.7** / **93.3** |

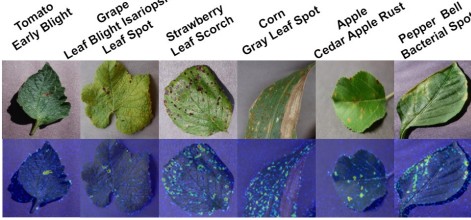

Table 2: Performance comparison of anomaly detection on VisA. Each cell reports max $F_1$-score (%) / image level AUROC (%) under N-shot settings.

Figure 7: Disease localization visualization on Plant with Disease dataset. Original images (top) and corresponding heat maps (bottom) demonstrate SD-LRT's ability to identify disease-affected regions across different plant leaf pathologies accurately.

**Low-shot regimes favor CLIP-based approaches.** That said, in extreme low-shot settings (*e.g.*, 1- or 2-shot), CLIP-based approaches such as LoCoOp and Tip-Adapter-F often outperform SD-LRT. For example, under the *Inter-12* setting with 1-shot supervision, MaPLe (**71.2**) significantly outperforms SD-LRT (**47.5**) in terms of OSR performance. We hypothesize that 1 or 2 training samples may be insufficient for SD-LRT to learn a reliable conditional likelihood in Eq. (7).

Overall, we consistently observe strong performance of the likelihood-based SD-LRT across different tasks, including the challenging industrial anomaly detection task in Table 2. SD-LRT also enables explainable classification as shown in Figure 7.

## 6 CONCLUSION

This work revisits the problem of open-set recognition (OSR) in the era of vision-language models (VLMs). We show that few-shot fine-tuning of VLMs remains necessary when the label granularity in downstream tasks diverges from the supervision seen during pre-training, in which case OSR remains challenging. To systematically investigate this, we introduce OpenVL-Bench, a comprehensive benchmark comprising 60 OSR tasks across diverse domains and hierarchical label structures. We evaluate two fine-tuning paradigms for VLMs: discriminative approaches that optimize for closed-set classification, and likelihood-based approaches that learn class-conditional densities using generative models. Our findings indicate that discriminative methods often misalign with standard OSR hardness metrics and lack robustness in open-set evaluation. In contrast, we show that likelihood-based methods become more tractable in the context of VLMs. Specifically, we propose a simple yet effective OSR method based on the likelihood-ratio test applied to diffusion-based VLMs, which achieves strong empirical performance. Overall, our study reaffirms the relevance of OSR in open-vocabulary settings and provides actionable insights for adapting VLMs toward robust open-world classification. Future work may explore improving the sample efficiency and inference speed of likelihood-based approaches, or addressing the inherent limitations of discriminative methods in open-set scenarios.

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
