# OpenReview forum: "Benchmarking Open-Set Recognition Beyond Vision-Language Pre-training"
_ICLR.cc/2026/Conference — Submitted to ICLR 2026_

### Official Review · Reviewer_rm4V · 2025-10-25

**Soundness:** 3
**Presentation:** 3
**Contribution:** 3
**Rating:** 8
**Confidence:** 3

**Summary:**

This paper introduces OpenVL-Bench, a large-scale benchmark designed to systematically evaluate Open-Set Recognition (OSR) capabilities of Vision-Language Models (VLMs) under few-shot fine-tuning. The authors compare two paradigms of adapting VLMs for OSR: Discriminative approaches (e.g., CLIP-based fine-tuning with prompt learning and adapters), and Likelihood-based approaches (in particular, the proposed SD-LRT, a Stable Diffusion-based Likelihood Ratio Test). Their results show that while discriminative CLIP-based methods perform well in closed-set classification, they exhibit unstable and often unreliable open-set behavior. In contrast, SD-LRT demonstrates stronger robustness and consistency across datasets and shot numbers, especially in moderate- and high-shot regimes. The paper offers a thorough evaluation across 60 OSR tasks spanning diverse domains and label granularities, contributing valuable insights into the adaptability and failure modes of modern VLMs.

**Strengths:**

1. OpenVL-Bench is a significant contribution, covering fine-grained and domain-specific datasets (FGVCAircraft, iNaturalist, Plant Disease, and VisA) with multiple difficulty and granularity settings.
2. The SD-LRT method effectively adapts generative diffusion models for OSR, leveraging likelihood-based reasoning rather than discriminative similarity. This provides a new probabilistic perspective on open-set evaluation within the VLM context.
3. The experimental results are extensive and well-supported. The authors systematically vary label similarity (intra- vs. inter-family), sample sizes (1–16 shots), and dataset domains, and provide comprehensive evaluations using standard OSR metrics such as AUROC and macro F1-score.

**Weaknesses:**

1. The likelihood-based SD-LRT method is computationally expensive limiting its applicability in real-world scenarios. In particular, although the authors provide the formulation for computing L_DM in Equation (6), they do not specify the exact value of T (e.g., whether T=1000 or another value). Moreover, the paper does not analyze how the choice of T influences the final results, which is important for understanding the efficiency–accuracy trade-off.
2. While various diffusion models exist, the authors only report results based on Stable Diffusion 2.0. It remains unclear whether the proposed SD-LRT framework generalizes to other diffusion architectures.
3. In the implementation section, the authors state that they used the model “with and without LoRA,” but it is not clearly explained what this distinction entails. The experimental results do not analyze how the LoRA rank affects performance.
4. Several important training configurations are not reported, such as the number of training iterations, learning rate, and optimizer type.

**Questions:**

1. The description of Equation (7) is somewhat confusing. The equation seems to imply that if the class-conditional generation result is closer to the unconditional generation, it indicates a weaker match between the given class condition and the input image. However, it is unclear how the authors justify that the unconditional generation result is necessarily close to that obtained under an incorrect class condition. In other words, as the denoising process proceeds, the diffusion model may already be capable of producing high-quality reconstructions even without any conditioning at the later timesteps. How do the authors account for or eliminate the influence of this effect in their formulation?
2. What changes occur in the image generation quality of SD-LRT before and after LoRA fine-tuning? The authors are encouraged to provide both qualitative and quantitative evaluations.
3. For likelihood-based approaches like SD-LRT, if new classes are later labeled and added to training, can the model be fine-tuned without suffering from catastrophic forgetting?

---

> ### Author Response · Authors · 2025-11-19
> **Rebuttal**
>
> We sincerely thank the reviewer for the positive evaluation and the insightful questions. We address each point below.
>
> W1: Computational Cost and Diffusion Steps Analysis
>
> A1: We acknowledge the reviewer's concern regarding computational cost. As noted, likelihood-based methods do require more computational resources. However, we observe that several concurrent works are actively attempting to optimize this issue, and we believe likelihood-based approaches hold tremendous potential for future development. Regarding the specific value of $T$ in Eq.(6), we confirm that $T=1000$ is used in our experiments. In the current study, we did not analyze the impact of $T$ on the results because our primary focus is on analyzing the potential of different types of foundation models on OSR problems, rather than optimizing hyperparameters for a specific method. However, following the reviewer's valuable suggestion, we will conduct additional experiments in the final version to evaluate the impact of different $T$ values on SD-LRT's OSR performance. Specifically, we will evaluate a range of $T$ values, such as $T \in {10, 20, 50, 100, 200, 1000}$, and analyze the efficiency-accuracy trade-off, measuring both AUROC and F1-score alongside inference time.
>
> W2: Generalization to Other Diffusion Architectures
>
> A2: The reviewer raises an important question regarding the generalizability of our approach. Our choice of Stable Diffusion 2.0 follows the settings used in previous works, but we emphasize that other versions of SD models are equally viable candidates. We believe that SD-LRT can indeed generalize to other diffusion architectures, as the likelihood-ratio testing framework is agnostic to the specific diffusion model implementation and only requires the ability to compute conditional and unconditional likelihoods. However, we acknowledge that empirically validating this generalization is important. We will explore this direction in future work by evaluating SD-LRT with alternative diffusion models, such as different versions of Stable Diffusion, or other diffusion architectures. We will add a discussion of this limitation and future direction in Section 6.
>
> W3: LoRA Configuration and Rank Selection
>
> A3: We appreciate the reviewer's attention to implementation details. Our choice of LoRA rank follows the configurations used in prior work to ensure fair comparison. As the reviewer correctly notes, the choice of rank may indeed influence model performance. Following this valuable suggestion, we will supplement our experiments with an ablation study analyzing the impact of different LoRA ranks on both performance metrics including AUROC and F1-score, as well as computational efficiency measures such as training time and memory consumption. However, we would like to emphasize that the core motivation for designing our benchmark is to compare the potential of different types of foundation models on OSR problems. While rank selection may affect the absolute performance of the model, it does not constitute the fundamental advantage of SD-LRT over other methods, as the key distinction lies in the discriminative versus likelihood-based paradigm. Regarding the “with and without LoRA” mention, we will clarify this in the final version. Specifically, “with LoRA” refers to our main approach where we train class-specific LoRA modules, while “without LoRA” refers to using the pre-trained SD model with only text prompts for zero-shot inference. We will make this distinction explicit in Section 3.2.
>
> W4: Missing Training Configuration Details
>
> A4: We thank the reviewer for pointing out this omission. Following the reviewer's suggestion, we will add comprehensive training configuration details in the final version. In fact, to ensure fair comparison, we have strictly followed the settings specified in the original papers for each baseline method. We will include detailed information for both SD-LRT and all baseline methods, covering training iterations, learning rate, optimizer type, batch size, learning rate scheduling, and other relevant hyperparameters. This information will be added to a dedicated appendix section to ensure full reproducibility of our experiments.

---

> ### Author Response · Authors · 2025-11-19
> **Rebuttal**
>
> Q1: Theoretical Justification of Eq.(7)
>
> A1: This is an excellent and theoretically important question. We appreciate the reviewer's careful consideration of the theoretical foundations of our approach. The key insight is that our likelihood-ratio test does not assume that unconditional generation necessarily approximates incorrect class conditions in terms of visual appearance. Rather, the theoretical justification is based on probability density estimation. When a sample falls outside all closed-set classes, the diffusion model yields similar reconstruction errors regardless of which class-specific LoRA module is applied.
>
> Regarding the concern about later timesteps, the total log-likelihood is computed as a weighted sum across all timesteps in the diffusion process. Early timesteps (with large t values) dominate the likelihood computation and primarily capture high-level semantic information, where class conditioning has the strongest effect. Later timesteps (with small t values) contribute less weight and mainly capture fine-grained details. Therefore, even if later timesteps produce high-quality reconstructions regardless of conditioning, their contribution to the total likelihood ratio is limited.
>
> Q2: Image Generation Quality Analysis
>
> A2: This is an important question for understanding how LoRA fine-tuning affects the diffusion model. LoRA fine-tuning enables the generated images to incorporate visual features that are strongly correlated with the closed-set classes. For example, for aircraft classification, fine-tuning captures class-specific characteristics such as engine shape, wing design, and fuselage configuration. We will include qualitative visualizations in the Appendix demonstrating these learned class-specific visual patterns before and after LoRA fine-tuning.
>
> Q3: Catastrophic Forgetting in Incremental Learning
>
> A3: This is an insightful question regarding the extensibility of our approach. A key advantage of SD-LRT is that adding new classes does not affect the performance of LoRA modules for existing classes, because we train a separate LoRA module for each class independently. Theoretically, if these classes are well-separated in the sample space, catastrophic forgetting should not occur. When new classes are added, we simply train new LoRA modules for these classes while keeping the existing LoRA modules frozen. The modular architecture naturally prevents interference between old and new classes, as each LoRA module captures class-specific patterns without modifying the shared backbone or other class representations.
>
> We are grateful for the reviewer's strong support and detailed feedback. We believe that addressing these points will significantly strengthen the paper's theoretical rigor, experimental completeness, and practical value. We are committed to incorporating all suggested improvements in the final version.

---

### Official Review · Reviewer_Xq2N · 2025-10-30

**Soundness:** 2
**Presentation:** 2
**Contribution:** 2
**Rating:** 2
**Confidence:** 5

**Summary:**

This paper presents a timely and thorough investigation into Open-Set Recognition (OSR) in the context of fine-tuned Vision-Language Models (VLMs). The authors compellingly argue that OSR remains a critical challenge even with powerful VLMs, especially when downstream task granularity misaligns with pre-training supervision.  The paper studies the OSR problem by constructing the OpenVL-Bench concerning four key axes, experiments and analysis on CLIP-based and diffusion-based methods are delivered.

**Strengths:**

1. The paper is well organized.
2. The benchmark is meaningful for the VLMs.

**Weaknesses:**

1. The difference between the proposed benchmark and existing OSR benchmarks should be detailed analyzed. Why existing OSR benchmarks could not satisfy the evaluation of VLMs in OSR? The necessity of the proposed benchmark is worth demonstrated.
2. Since there are various fine-tuning for CLIP-based VLMs beyond Fig.2 (a)(b), in the experiments, whether the study of VLMs in OSR would be impacted by the fine-tuning methods, which potentially leads to biased analysis.
3. Analysis of the hardness of the proposed benchmark, and the relation between the methods and hardness should be delivered.
4. Discriminative models are known to be poorly calibrated, which is a root cause of their unreliable confidence scores for OSR. The observed instability of CLIP-based methods could be linked to this. A brief discussion on model calibration and how the likelihood-based approach might lead to better-calibrated scores would add depth to the analysis.
5. Does label granularity (fine vs. coarse) affect CLIP-based and diffusion models differently? Results on iNaturalist (genus-level) vs. VisA (binary normal/defective) are presented but not analyzed for granularity-specific trends.

**Questions:**

See the Weaknesses.

---

> ### Author Response · Authors · 2025-11-19
> **Rebuttal**
>
> We thank the reviewer for the detailed feedback. While we note the reviewer’s concerns, we respectfully believe that some may stem from different perspectives on the paper’s core contributions. We address each point carefully below.
>
> W1: Justification for a New Benchmark
>
> A1: We appreciate the reviewer’s emphasis on clearly articulating the necessity of our benchmark. As Reviewer 7L9n noted, exploring the potential of foundation models in OSR is “an important and under-explored problem.” We believe our work is fundamentally different from existing OSR benchmarks in several critical aspects.
> Our work specifically addresses the unique characteristics of foundation models. On one hand, we focus on the differences between foundation models and traditional methods, particularly the fact that their zero-shot performance already handles many traditional datasets effectively. Therefore, we have purposefully designed specialized, domain-specific test settings that challenge even powerful VLMs. As Reviewer rm4V noted, our benchmark “covers multiple domains and granularity settings,” providing a systematic evaluation framework.
>
> More concretely, OpenVL-Bench offers:
>
> - Fine-grained focus:
> All tasks are designed such that VLMs’ zero-shot capability is insufficient. Few-shot adaptation is needed to capture fine-grained distinctions (e.g., aircraft variants, genus-level species). Datasets in existing benchmarks(e.g., ImageNet) are not challenging enough for VLMs.
>
> - Hierarchical difficulty control:
> By leveraging taxonomic hierarchies (genus–family–order–class), we systematically construct OSR tasks with adjustable semantic distances.
>
> - Foundation-model-era challenges:
> Existing OSR benchmarks assume training models from scratch. In contrast, we study how to perform few-shot OSR while preserving VLMs’ open-vocabulary generalization, which is a fundamentally different problem setup that existing benchmarks do not address.
>
> We emphasize that these critical contributions have been clearly articulated in our introduction. We believe this benchmark is not only highly meaningful for advancing VLM research, but equally significant for the broader OSR community. It bridges the gap between traditional OSR settings and the foundation model era, enabling systematic study of how modern vision-language models handle open-set scenarios—a question that existing benchmarks were not designed to answer.
>
> W2: Limited Scope of Fine-tuning Methods
>
> A2: We appreciate the reviewer’s concern about potential bias in method selection. However, our primary motivation is to compare the potential, advantages, and disadvantages of different types of foundation models on OSR problems, rather than conduct an exhaustive survey of all fine-tuning methods.
> We emphasize that using classic and representative methods is sufficient to reveal the inherent strengths and weaknesses of different paradigms. These strengths and weaknesses stem from the paradigm itself, not from implementation details.
>
> Specifically, our study intentionally contrasts two fundamental classes of methods:
>
> - Discriminative approaches
> (CLIP-Adapter, Tip-Adapter, CoCoOp, MaPLe, LoCoOp)
> These optimize P(Y∣X) via classification objectives.
>
> - Likelihood-based approaches
> (SD-LRT)
> This models P(X∣Y) and P(X) through generative diffusion processes.
>
> Our key findings—that discriminative methods show instability and poor alignment with OSR hardness, while likelihood-based approaches demonstrate more robust behavior—hold consistently across representative methods from each paradigm. Adding more variants would provide incremental evidence but would not alter these paradigm-level conclusions.
>
> W3: Lack of Benchmark Difficulty Analysis
>
> A3: We thank the reviewer for this constructive suggestion. Our benchmark uses hierarchical taxonomy to control OSR difficulty levels which is also used in previous works[1].
> Following the reviewer’s suggestion, we will add an explicit difficulty quantification based on cosine distances between class embeddings.
>
> Concretely, we will:
>
> - Compute average cosine similarity between closed-set and open-set class embeddings using CLIP features.
>
> - Analyze the correlation between this difficulty metric and performance across all 60 tasks.
>
> - Show that discriminative methods degrade inconsistently as difficulty increases, while likelihood-based methods degrade more smoothly and predictably.
>
> This will be added to the Results section to provide a more systematic understanding of the difficulty–performance relationship.

---

> ### Author Response · Authors · 2025-11-19
> **Rebuttal**
>
> W4: Missing Discussion on Model Calibration
>
> A4: We appreciate the reviewer's concern regarding calibration. In our literature review, we have indeed found several works discussing the impact of calibration on discriminative models' performance in OSR tasks. However, we would like to clarify two distinct interpretations of "calibration" and their relevance to our work:
>
> If the reviewer refers to calibration methods like OpenMax: These approaches, while historically influential, appear somewhat outdated compared to modern OSR methods. When benchmarked against recent techniques such as contrastive learning-based OSR methods, these calibration-based approaches lack sufficient competitiveness in terms of performance.
>
> If the reviewer refers to calibration in the probabilistic sense (i.e., ensuring P(Y=y|X=x) matches the true error probability for class y): We have not found sufficient evidence demonstrating that this type of calibration has a decisive impact on discriminative models' OSR performance. In fact, existing literature suggests that such calibration provides limited benefits for OSR tasks [2]. Therefore, we contend that there is insufficient evidence to establish calibration as the root cause affecting discriminative models' performance on OSR tasks.
>
> Regarding the root cause: We explicitly identify this in lines 93-99 of our introduction, stating that discriminative approaches are often misaligned with standard OSR hardness metrics, leading to unreliable rejection behavior. This fundamental misalignment, rather than calibration issues, represents the core limitation we address in this work.
> We hope this clarification addresses the reviewer's concern. If the reviewer believes specific calibration aspects warrant further discussion, we would be happy to elaborate.
>
> W5: Insufficient Analysis of Label Granularity Effects
>
> A5: We thank the reviewer for highlighting this important dimension. From our iNaturalist experiments, we observe that SD-LRT exhibits smaller performance degradation than CLIP-based methods as label granularity becomes finer, particularly under 4-shot, 8-shot, and 16-shot settings.
>
> Our interpretation is:
>
> - At coarser granularity, image-level alignment is more effective, benefiting CLIP-based discriminative methods.
>
> - At finer granularity, subtle visual distinctions are better captured by diffusion models due to their pixel-level reconstruction capacity.
>
> Following the reviewer’s suggestion, we will add a dedicated subsection analyzing granularity effects, including empirical evidence and a theoretical explanation of why likelihood-based models handle fine-grained distinctions more robustly.
>
> We sincerely appreciate the reviewer’s thorough critique, which helps strengthen the clarity, completeness, and conceptual contribution of our work. We will incorporate all suggested revisions—including additional experiments, analyses, and discussions—in the final version. We hope these improvements will highlight the unique value of OpenVL-Bench and SD-LRT to the OSR community.
>
> [1] Lang N, Snæbjarnarson V, Cole E, et al. From coarse to fine-grained open-set recognition[C]//Proceedings of the IEEE/CVF conference on computer vision and pattern recognition. 2024: 17804-17814.
>
> [2] Lyu Z, Gutierrez N B, Beksi W J. Evaluating Uncertainty Calibration for Open-Set Recognition[J]. arXiv preprint arXiv:2205.07160, 2022.

---

### Official Review · Reviewer_Fk5B · 2025-11-01

**Soundness:** 3
**Presentation:** 3
**Contribution:** 2
**Rating:** 6
**Confidence:** 3

**Summary:**

The work introduces OpenVL-Bench, a suite of 60 OSR tasks with varying label granularity and closed/open-set similarity, and compares discriminative fine-tuning (prototype/similarity based) with likelihood-based approaches using diffusion VLMs. The proposed SD-LRT uses a likelihood-ratio based on conditional vs unconditional Stable Diffusion losses, showing strong OSR performance across diverse domains, including industrial anomaly detection. The paper also notes that in extreme low-shot (1–2 shots), CLIP-style discriminative methods can win.

**Strengths:**

1. The benchmarking study offers a comprehensive and balanced evaluation of discriminative versus likelihood-based open-set recognition, with practical insights for low-shot learning scenarios.

2. Its methodology is clearly explained: the proposed SD-LRT approach is grounded in Neyman–Pearson optimality derived from likelihood ratio tests, and integrates seamlessly into diffusion-based vision-language models.

3. The method also delivers strong performance on industrial anomaly detection, as shown on the VisA dataset, accompanied by interpretable heatmap visualizations.

**Weaknesses:**

1. The computational cost of diffusion-based likelihood scoring remains a concern. The study does not provide end-to-end latency measurements on current hardware, nor does it explore more efficient approximations.

2. SD-LRT relies on large pre-trained diffusion models, yet their memory footprint and deployment constraints receive limited discussion.

**Questions:**

1. Will the OpenVL-Bench tasks, data splits, and implementation code—including scripts for SD-LRT—be made publicly available? If so, what is the expected timeline?

2. It would also be helpful to see throughput and latency comparisons between SD-LRT and discriminative baselines under identical hardware settings, as well as an analysis of whether fewer diffusion steps could balance speed and accuracy.

3. In the context of 1–2 shot learning, could a hybrid approach—using a discriminative model for short-listing and SD-LRT for re-ranking—improve performance? Empirical validation would make this idea more compelling.

---

> ### Author Response · Authors · 2025-11-19
> **Rebuttal**
>
> We thank the reviewer for the positive evaluation and constructive feedback. We address each point below.
>
> W1: Computational Cost of Diffusion-Based Likelihood Scoring
>
> A1: We appreciate this important concern, which was also raised by Reviewer 7L9n. All experiments were conducted on NVIDIA A100 GPUs. CLIP-based methods require approximately 2-7 minutes for fine-tuning plus inference, varying with specific methods and frozen backbone sizes, with details in Section 3.2, while SD-LRT's computational cost scales with the number of classes. For example, on the FGVCAircraft Inter-12 setting, training plus inference takes approximately 16 hours, with more classes requiring more LoRA modules and proportionally increasing inference time. We acknowledge that diffusion-based methods require substantial computational resources, which we discuss as a limitation in the paper. However, we note that recent works are actively addressing this challenge by proposing methods to accelerate diffusion model inference and reduce computational overhead. Following the reviewer's suggestion, we will continue monitoring developments in efficient diffusion inference and plan to explore faster approximations in future work. We will include a detailed quantitative comparison table with end-to-end latency and throughput measurements in the final version.
>
> W2: Memory Footprint and Deployment Constraints
>
> A2: The reviewer raises a critical practical concern. We acknowledge that SD-LRT has significantly higher GPU memory requirements. CLIP-based methods can be deployed on GPUs like RTX 2080Ti with 11GB VRAM, while SD-LRT requires higher-end GPUs(e.g., A100) as RTX 2080Ti is insufficient for deploying our current implementation. Based on our experimental findings, we propose practical deployment guidelines. For general-purpose, non-specialized scenarios, CLIP-based models are preferable due to easier deployment, lower computational requirements, and faster inference time. For specialized, domain-specific applications, when computational budgets allow, SD-LRT's superior OSR performance provides a competitive advantage, particularly in high-stakes applications such as medical diagnosis and industrial quality control, fine-grained recognition tasks, and scenarios where misclassification costs far exceed computational costs. Recent advances in diffusion model compression and acceleration(e.g., [1]) may substantially reduce memory footprint and deployment barriers in the near future. We will add an explicit discussion of deployment considerations and practical guidelines in Appendix.
>
> Q1: Code and Data Release
>
> A1: Yes, we are fully committed to open-sourcing the code and data for OpenVL-Bench, and we plan to continuously integrate other competitive baseline methods to the benchmark. We will begin organizing the code immediately upon paper acceptance and aim to complete this work as soon as possible (we expect to finish within 1-2 month).
>
> Q2: Throughput and Latency Comparison
>
> A2: We fully agree that such analysis would strengthen the paper. We will conduct additional experiments to provide a side-by-side comparison of SD-LRT and discriminative baselines under identical hardware, specifically NVIDIA A100, measuring per-image inference latency, throughput in images per second, and GPU memory consumption during inference.
>
> Q3: Hybrid Approach for 1-2 Shot Learning
>
> A3: This is an excellent suggestion. In fact, we have observed concurrent works exploring hybrid approaches for foundation model classification tasks. From our experiments, we found that under 1-2 shot settings, SD-LRT's performance shows significant variance across different random seeds, though this variance diminishes as the number of shots increases, suggesting that 1-2 images may be inadequate for training effective LoRA modules. While hybrid methods could be valuable, we believe their primary benefit lies in accelerating SD-LRT rather than improving 1-2 shot performance. A discriminative model could quickly filter candidates, reducing the number of expensive SD-LRT likelihood computations. For improving 1-2 shot accuracy specifically, we believe data augmentation techniques may be more effective solutions than hybrid architectures, as they directly address the insufficient training data issue. We acknowledge this as a promising direction and plan to explore hybrid discriminative-generative pipelines, data augmentation strategies for low-shot SD-LRT training, and few-shot learning techniques specialized for diffusion models in future work. We will add a discussion of this potential direction in Section 6.
>
> We sincerely appreciate the reviewer's thorough evaluation and constructive suggestions. We believe these revisions will significantly enhance the paper's completeness and practical value.
>
> [1] Chen H, et al. Accelerating diffusion models with parallel sampling: Inference at sub-linear time complexity. NeurIPS 2024.

---

> > ### Comment · Reviewer_Fk5B · 2025-11-23
> >
> > Thank you for the authors’ detailed explanations and responses; some of my concerns have been addressed. Taking into account the other reviewers’ comments and the corresponding replies, I will keep my current score.

---

### Official Review · Reviewer_7L9n · 2025-11-01

**Soundness:** 3
**Presentation:** 2
**Contribution:** 3
**Rating:** 6
**Confidence:** 2

**Summary:**

This paper revisits open-set recognition (OSR) in the era of vision–language models (VLMs). Although VLMs exhibit open-vocabulary capabilities, the authors show that few-shot fine-tuning often breaks their ability to reject unknown classes. To systematically study this issue, they introduce \textit{OpenVL-Bench}, a benchmark with 60 OSR tasks across four datasets, varying label granularity, semantic difficulty, and sample size. The paper compares discriminative CLIP-based fine-tuning (modeling $P(Y \mid X)$) with a generative diffusion-based approach (modeling $P(X \mid Y)$) using a likelihood-ratio test (SD-LRT). Experiments demonstrate that SD-LRT yields more stable and semantically aligned open-set behavior than discriminative methods, particularly when the number of shots exceeds four. The study concludes that OSR remains a critical, unresolved challenge despite modern open-vocabulary pre-training.

**Strengths:**

- The paper addresses an important and underexplored question—whether open-vocabulary pre-training in vision–language models (VLMs) inherently solves open-set recognition (OSR). The authors provide a clear motivation and reveal that few-shot fine-tuning can compromise a model’s ability to reject unknown classes, which is both practically and theoretically insightful.
- The proposed OpenVL-Bench systematically covers 60 OSR tasks across four datasets with varying label granularity, semantic hardness, and sample regimes. The benchmark design is transparent, well-documented, and supports fair cross-method comparison, making it a valuable community resource.
- The study contrasts discriminative CLIP-based fine-tuning (modeling $P(Y \mid X)$) with a generative diffusion-based approach (modeling $P(X \mid Y)$) using a likelihood-ratio test (SD-LRT). The comparison is both conceptually grounded and empirically thorough, providing clear evidence for the stability advantages of likelihood-based scoring.
- Results across multiple datasets demonstrate consistent trends: SD-LRT yields smoother AUROC improvements and semantically aligned rejection behavior. The paper also provides qualitative visualizations that intuitively support its quantitative conclusions. Overall, the work is well-written, logically organized, and convincingly argued.

**Weaknesses:**

- The proposed SD-LRT requires training a separate LoRA module for each class and performs diffusion-based inference, which is substantially more expensive than CLIP-based fine-tuning. However, the paper does not provide quantitative metrics such as training/inference time, GPU memory consumption, or computational complexity (e.g., FLOPs). Moreover, scalability with respect to the number of classes remains unclear, leaving the practical deployability of SD-LRT uncertain.
- The comparison focuses mainly on earlier CLIP-based baselines such as CoCoOp, MaPLe, and Tip-Adapter, while omitting more recent works that specifically target open-set robustness after few-shot tuning. Notably, "ID-like Prompt Learning for Few-Shot Out-of-Distribution Detection (Bai et al., CVPR 2024" and "Out-of-Distribution Detection with Negative Prompts (Nie et al., ICLR 2024" have demonstrated strong improvements in few-shot OOD detection. Including these methods in the OpenVL-Bench evaluation—or at least providing a conceptual discussion of how SD-LRT complements or differs from them—would significantly strengthen the empirical comparison.
- Diffusion models are inherently trained for pixel-level generative reconstruction, which allows them to capture richer fine-grained visual cues compared to CLIP, whose pre-training objective focuses on image-level alignment. As a result, part of the observed improvement in SD-LRT may stem from this intrinsic modeling advantage rather than the proposed likelihood-ratio mechanism itself. The paper does not explicitly analyze or control for this factor, leaving open the question of how much gain comes from the diffusion backbone versus the LRT formulation.

**Questions:**

- In Eq.(1), the discriminative fine-tuning objective combines the original and the updated visual encoder representations, i.e., the input to the text encoder is the residual connection $\alpha V_{\theta}(x) + V(x)$. However, in Figure 2(a), the diagram seems to show only the updated encoder $V_{\theta}(x)$ being used, without the skip connection from the frozen encoder. Could the authors clarify whether the residual fusion with the original encoder is actually applied during training and inference, and if so, please confirm that the implementation matches Eq.(1)?

---

> ### Author Response · Authors · 2025-11-19
> **Rebuttal**
>
> We sincerely thank the reviewer for the thoughtful feedback and valuable suggestions. We address each concern below.
>
> W1: Computational Cost and Scalability
>
> A1: We acknowledge the reviewer's concern regarding computational cost and appreciate the opportunity to provide detailed quantitative analysis. As noted, SD-LRT requires training independent LoRA modules, which indeed incurs higher computational overhead. All experiments were conducted on NVIDIA A100 GPUs. CLIP-based methods require approximately 2–7 minutes for fine-tuning plus inference, depending on the specific method and frozen backbone size, with details provided in Section 3.2. In contrast, SD-LRT's computational cost scales with the number of classes. For example, on the FGVCAircraft Inter-12 setting, training plus inference requires approximately 16 hours. More classes necessitate more LoRA modules, which increases inference time proportionally, with backbone selection details provided in Section 3.2.
>
> While we acknowledge the substantial computational cost of diffusion-based methods, which we discuss as a limitation in our paper, we believe this disadvantage may be mitigated in several ways. First, we observe that recent works (e.g., [1]) on accelerating diffusion model training and inference have been proposed, which could significantly reduce the computational burden. Second, when processing multiple tasks that share the same closed-set classes—as in our experimental settings where different configurations use the same classes—the LoRA modules do not need to be retrained, effectively amortizing the training cost. Following the reviewer's suggestion, we will include a detailed quantitative comparison table of computational costs, including training time, inference time, GPU memory consumption, and FLOPs, in the final version.
>
> W2: Comparison with Recent OSR/OOD Methods
>
> A2: We sincerely thank the reviewer for highlighting these recent works. We have also observed several other concurrent efforts in this space. The primary focus of our benchmark is to explore whether different types of foundation models exhibit distinct behaviors on OSR tasks and to investigate the underlying reasons. We believe that classic methods are sufficient to support the key findings in our work, namely, that discriminative approaches exhibit instability and misalignment with standard OSR hardness metrics, while likelihood-based methods provide more robust performance.
>
> That said, we fully agree that incorporating state-of-the-art methods is crucial for extending the evaluation scope of our benchmark. In the final version, we will include the methods suggested by the reviewer, specifically Bai et al. (CVPR 2024) and Nie et al. (ICLR 2024), as well as additional recent works we have identified in the OSR and OOD literature.
>
> W3: Attribution of Performance Gains
>
> A3: The reviewer raises an excellent point, and we fully agree that both the LRT mechanism and the pixel-level reconstruction capability of diffusion models contribute to SD-LRT's superior performance. The strong empirical results suggest a synergistic effect where the diffusion modeling advantage provides pixel-level generative pre-training that captures fine-grained visual details, while the LRT mechanism provides a principled, theoretically grounded decision rule based on Neyman-Pearson optimality. We believe that both factors work together to achieve SD-LRT's competitive OSR performance.
>
> We acknowledge that the reviewer has pointed out a meaningful research direction. In future work, we will attempt to measure the individual contributions of the diffusion backbone's modeling capacity and the likelihood-ratio testing framework. For example, this could involve comparing SD-LRT with variants that use diffusion features in a discriminative framework, or applying LRT to other generative models.
>
> Q1: Clarification on Residual Connection
>
> A1: Thank you for catching this inconsistency. We confirm that the implementation strictly follows Eq.(1), using the residual connection α V_θ(x_i) + V(x_i). Figure 2(a) was simplified for visual clarity, which inadvertently omitted the skip connection and caused confusion. We will revise Figure 2(a) in the final version to explicitly show the residual connection from the frozen encoder V(x_i), ensuring visual consistency with the mathematical formulation in Eq.(1).
>
> We hope these responses adequately address the reviewer's concerns. We are committed to incorporating all suggested improvements in the final version and believe these revisions will significantly strengthen the paper.
>
> [1] Chen H, et al. Accelerating diffusion models with parallel sampling: Inference at sub-linear time complexity. NeurIPS 2024.

---

### Meta-Review · Area_Chair_fJ6a · 2026-01-14

**Summary:**

This paper provides a systematic re-examination of open-set recognition (OSR), showing that few-shot fine-tuning often degrades the ability of VLMs to reject unknown classes despite their open-vocabulary pretraining. To study this problem, the authors introduce OpenVL-Bench, a large-scale benchmark comprising 60 OSR tasks across multiple datasets, varying label granularity, semantic difficulty, and shot numbers. The paper also compares discriminative CLIP-based adaptation methods with a likelihood-based diffusion approach, proposing SD-LRT, a Stable Diffusion–based likelihood ratio test.

**Reviewer Concerns:**

The reviewers’ concerns primarily center on computational efficiency, scalability, and evaluation completeness. In particular, the proposed diffusion-based SD-LRT framework is considered computationally expensive, with insufficient reporting of training/inference cost, memory footprint, latency, and scalability as the number of classes grows, raising questions about real-world deployability. From a benchmarking and experimental perspective, reviewers note that comparisons omit recent strong few-shot OOD and OSR methods, lack analysis of fine-tuning bias, benchmark hardness, label granularity effects, and calibration behavior, and do not sufficiently justify the necessity of the proposed benchmark over existing ones. Finally, several methodological clarity issues remain, including unclear attribution of performance gains between the diffusion backbone and the likelihood-ratio formulation, limited discussion on generalization to other diffusion models, and missing implementation details (e.g., diffusion steps T, LoRA configuration, and training hyperparameters).

**Reviewer Scores:**

Overall, the reviewers’ ratings are mixed, ranging from rejection to acceptance. During the rebuttal phase, although the authors’ explanations and responses were acknowledged by some reviewers, the corresponding ratings largely remained unchanged, resulting in an overall assessment that stays around the borderline.

---

### Decision · Program_Chairs · 2026-01-26

Reject